# Knowledge Is Power: Utilizing Human-Centered Design Principles with People Living with Dementia to Co-Design a Resource and Share Knowledge with Peers

**DOI:** 10.3390/ijerph20206937

**Published:** 2023-10-18

**Authors:** Jennifer Rhiannon Roberts, Catrin Hedd Jones, Gill Windle

**Affiliations:** DSDC Wales Research Centre, School of Health Sciences, Bangor University, Ardudwy, Normal Site, Bangor LL57 2PZ, UK; c.h.jones@bangor.ac.uk (C.H.J.); g.windle@bangor.ac.uk (G.W.)

**Keywords:** dementia, young onset, co-design, peer support, lived experience, resilience, wellbeing, diagnosis, post-diagnosis support, public health

## Abstract

This paper describes the process used by a group of people living with young-onset dementia to inform the development and delivery of a post-diagnosis peer guide. It draws on the four stages of human-centered design and applies them in a new context of supporting resilience for people following a diagnosis of dementia. (1) Discover: The group discussed in-depth their perspectives on what it takes to be resilient while living with dementia and how this can be maintained. (2) Define: The group decided to collate practical information and knowledge based on their personal experiences into a booklet to support the resilience of others following a diagnosis of dementia. (3) Develop: The booklet was designed and developed together with input from other people living with dementia, facilitated by the authors. (4) Deliver: The group guided the professional production of the booklet ‘Knowledge is Power’. Over 8000 copies have been distributed to memory clinics, post-diagnostic support organizations and people living with dementia across Wales. A bilingual English–Scottish Gaelic adaptation and an adaptation for people in England have since been developed. The success of ‘Knowledge is Power’ highlights the importance of working alongside people with dementia to share knowledge and support their resilience.

## 1. Introduction

Best-practice recommendations by the World Health Organization [1] emphasize the importance of empowering people living with dementia in being involved in advocacy, service provision and research, and the National Institute for Health and Care Excellence (NICE) guidelines suggest encouraging and enabling people living with dementia to give their own thoughts and opinions about their care [2], as do national dementia strategies, e.g., [3,4]. However, opportunities to participate in designing services and resources aimed to support others living with dementia are still limited [5,6]. A review by Bombard et al. [5] on the engagement of patients to improve quality of care only included one study by Tooke [7], which involved asking people living with dementia to review pieces of work being undertaken by Alzheimer’s Society staff.

More recently, Dupuis et al. [6] took an action research approach to create and share information with others on dementia diagnosis and what it entails by developing an online self-management resource together with people with dementia and a range of professionals. Similarly, Pipon-Young et al. [8] described an action research study in which people living with young-onset dementia decided on the action of creating a leaflet to share with others receiving a diagnosis of dementia due to their own feelings of receiving inadequate support and information. Action research involves the exploration of a problem by those directly affected and subsequently turning this understanding into action to improve the lives of others potentially facing the same issues [9]. Dupuis et al. [6] emphasize the importance of including people with dementia in developing the resource as it ensures that the decisions made during the process are relevant to those with the lived experience.

Human-centered design, co-design, design thinking, and user-centered design are terms often used interchangeably to describe collaboration between professionals and people with lived experience as equal partners in developing a product or service. It is an innovative approach to developing design solutions which focusses on the needs of the users in all stages of the design. It draws some parallels with action research, but rather than being a research process, it is a direct application towards the design of a product using techniques which “communicate, interact, empathize and stimulate the people involved, obtaining an understanding of their needs, desires and experiences” [10] (p. 610). It moves away from the type of approach that assumes that service providers are best placed to know about the problem or indeed the best solution [11], and instead relies on the involvement of ‘end-users’ throughout the process, thus bringing a wealth of resources, knowledge, and lived experience that would otherwise not be possible. Several frameworks demonstrate the human-centered design approach (e.g., ‘IDEO’—inspiration, ideation, implementation [12]; ‘d.school’—empathize, define, ideate, prototype, test [13]; or the ‘Double Diamond’—discover, define, develop, deliver [14]). The number and names of stages within the design process differ but their key principles are the same: human-centered design places people with lived experience at the heart of the process from start to finish to ensure that the resulting resource or product is suitable.

Human-centered design has been used in the context of health and with people living with various conditions, such as developing a tablet/computer-based self-management support for diabetes [15], virtual reality-based therapy for post-traumatic stress disorder [16], and video-based health education [17]. In relation to dementia, this type of design process has been used to create a web-based psychoeducational program to reduce stress in carers of people living with Alzheimer’s disease [18] and a mobile application (‘RefineMind’) for people living with dementia and their carers [19], and a similar approach was used by Black et al. [20], who describe the co-design of a printed information handbook for carers of people with dementia.

This article describes the human-centered design process adopted by people living with young-onset dementia in Wales (UK) to co-design guides for others following a diagnosis of dementia, facilitated by academics. The booklets are based on knowledge gained and shared amongst peers through their own experiences while living with dementia. The booklet production was inspired by a discussion about the concept of resilience, where people with dementia and their carers used the phrase “knowledge is power” to describe how learning about living with dementia was important for their resilience.

To our knowledge, this article is the first to use human-centered design principles in the development of a peer-to-peer resource by and for people living with dementia.

## 2. Materials and Methods

### Participants

The Caban group includes people living with young-onset dementia and carers who have been working closely with academics and students since 2017 and aims to share the experience of people living with dementia and increase understanding. The group emerged from a regional dementia network established by the authors in 2016 that included people affected personally by dementia and a range of people working to support those diagnosed with dementia. The North Wales Dementia Network was established through an Impact Acceleration Award by the Economic and Social Research Council (ESRC). Network meetings included people living with dementia who were members of DEEP (Dementia Engagement and Empowerment Project; dementiavoices.org.uk), and the authors asked for their input into a consultation with health students at the university. The response from students to the opportunity to openly meet and talk to a person living with dementia inspired the establishment of a new DEEP group of ‘dementia educators’—the Caban Group. The group supports the work of the university through educating, discussing research, and collaborating with staff on raising awareness both within the university and at national events. The Caban group members (*n* = 7) driving this work were predominantly people with dementia, living with Alzheimer’s disease (*n* = 3) or a combination of both Alzheimer’s disease and vascular dementia (*n* = 2). All members apart from one were diagnosed under the age of 60. Table 1 provides the demographic information.

The development of this resource was led by the Caban group, with support from academics. This co-design work was not defined as research according to the Health Research Authority decision tool; however, good practice was adopted throughout. Members of the group provided written consent for submission of this paper and approved the content of the submitted manuscript in order to share the process and examples of good practice demonstrated in this work. The Chair of the School Research Ethics Committee was notified and a confirmatory statement from the chair provided.

To gain feedback about the booklet, a request for feedback using a short survey on Microsoft Forms was sent out via email and social media, open to anyone who had used or seen the booklet and wished to provide their opinions and recommendations. A statement was included at the beginning of the form notifying people that anonymous comments may be used to share feedback on the booklet. The feedback was collated by the lead author and all comments were considered by the group.

## 3. Design and Phases: Overview

This work draws on the human-centric principles of the UK Design Council’s ‘Double Diamond’ [14], adopting the following four phases: discover, define, develop and deliver, and applying it in a new context to inform the development and delivery of a new resource aimed at supporting resilience in people living with dementia. Table 2 summarizes how each stage was applied in this work.

### 3.1. Discover

The Caban group had been meeting with academics and students to share their expertise. A 2 h stakeholder engagement meeting with the authors took place in July 2019 in which the group discussed their perspectives on what it takes to be resilient while living with dementia. The discussion revealed many useful insights regarding adapting and managing life with dementia, and these are presented together with evidence from other people living with dementia by Windle et al. [21]. A combination of psychological strengths; practical approaches to adapting to life with dementia; continuing with hobbies, interests and activities; strong relationships with family and friends; peer support and education; participating in community activities; and support from healthcare professionals were resources deemed important for resilience in people with dementia.

A key theme in the discussions around threats to resilience was that bleak predictions accompanying the diagnosis of dementia can have a detrimental impact on a person’s resilience. Members of the group reported negative experiences, such as “At the beginning you are told you can’t do anything; it makes you feel useless” (Caban member). Reflecting on the diagnosis, the group described the information that they had received to be “too much” and “overwhelming”. The group agreed that “if you get too much information, you tend not to read it”. They also discussed the need for resources to be provided with the right amount of information and at the right time because:


*“When you find out all about your illness, and you have that information, knowledge is power. And then you can change your life; and once you’ve changed your life to suit your illness, then you can get on with it and live”*
(Caban member)

### 3.2. Define

As a result of this in-depth exploration of resilience, and after the meeting in July 2019, the group proposed to the authors that they wished to develop a resource, specifically a booklet, sharing practical information and knowledge based on their personal experiences and discussions with peers to support resilience following a diagnosis of dementia.


*“If people can get this [booklet] early enough… it would help them to maintain rather than to fall back and have to then find resilience”*
(Caban member)

The group wished for the printed booklet to benefit and maintain the resilience of people recently diagnosed with dementia in Wales. Wales is a largely rural country, a factor known to impede access to services, support, and adequate signposting post-diagnosis [22,23]. The Welsh Parliament (Y Senedd) has devolved responsibility for health and social care provision and a National Action Plan on Dementia [4]; therefore, the group decided that developing a booklet applicable for people living with dementia within Wales would be the most useful approach. They also specified that it should be available in Welsh and English, the official languages in Wales since 2011.

### 3.3. Develop

The group discussed the type and amount of information to include, and the consensus was to provide details of support available to the person who received the diagnosis and their unpaid carers in Wales through, for example, authorities and service providers. Caban members and facilitators invited input from other people living with dementia and their supporters and through existing DEEP groups in Wales. Requests for information focused on asking members to share “things I have found out along the way that other people living with dementia need to know” (the words of a Caban group member). Examples given to guide input included tips that would make life easier or less expensive, such as entitlements to payments or reductions, services that charities/companies are offering, and the types of health professional they may encounter and what their roles are. Only information applicable Wales-wide was to be included to ensure that the booklet could be relevant to anyone living with dementia in Wales and that it would contain a manageable amount of information.

Requests were presented at a DEEP United group meeting, in emails and by posting on the North and Mid-Wales Dementia Network Facebook groups, and further requests for submissions were also sent through the DEEP newsletter and website. A total of 47 people engaged in the exercise over the period of one month (Caban group (*n* = 7); other DEEP group attendees in Wales (*n* = 40)). Content was submitted using platforms the members were familiar with, including notes in meetings, emails, and Zoom [24] and Facebook Messenger [25]. COVID-19 restrictions prohibited in-person meetings for most of the process. Where required, members were supported to develop new video conferencing skills by JR through working together on the telephone to set up and use Zoom [24] to ensure that they were able to remain fully involved in the development of the booklet. Tablets were also purchased for use by those without access to other devices. The collated information was initially grouped thematically by JR, then shared with the Caban group for discussion using ‘screen share’ on the preferred videoconferencing software of the group, Zoom [24]. An iterative process of discussion, feedback and editing then refined the content of the booklet. During meetings, notes were taken by JR and, where possible, real-time edits were made while using the ‘screen share’ function. Drafts were prepared following online meetings and ‘group chats’ (Facebook Messenger [25]) over a period of 3 months, with amendments made at each stage to ensure that members were satisfied with the content of the booklet. Drafts of sections were prepared as a group, and, where necessary, JR would find further information about sections. The Caban group would then decide how much of this information should be included and how it should be worded. Zoom [24] meetings were arranged when necessary, but on average every two weeks. However, JR and the Caban group were in regular contact throughout the process.

DEEP guidelines were used throughout the making of the booklet to ensure that the written information was dementia friendly [26], and Caban members guided decisions around the design and appearance of the booklet, including size, font size, color schemes, layout, images, personal messages/tips to convey and the name of the booklet. The A5 size was appropriate for accessibility, in that it would be easy to put in a bag. A bilingual ‘tilt and turn’ booklet design, where the reader flips the booklet to read the other language, would enable readers to choose which language they would prefer to access the information. Once members were happy with the English content, a Welsh translation was prepared by the authors and professional translators within the same organization.

Finally, feedback on an initial draft booklet was sought from a wide range of sources, including people living with dementia, carers and professionals (e.g., local council and National Health Service staff and the Wales lead Consultant for Allied Health Professionals for Dementia). The outcome was the first bilingual booklet consolidating information considered most important to share with others following a diagnosis of dementia in Wales. It was named ‘Knowledge is Power/Mae Pŵer mewn Gwybodaeth’ by the group, which is a direct quote from one of the members in relation to their perspective of resilience in dementia. It included the following sections: Some advice from us to you; Benefits and allowances; Specifically for carers; Respite or breaks; Discounts you may be entitled to; Travel and mobility; Access assistance; Schemes and campaigns; Legal considerations; Support you may come across; Useful websites and phone numbers.

### 3.4. Deliver

The booklet was professionally produced and printed by a graphic designer, with guidance from the group members, through JR. Funding from DEEP and Health and Care Research Wales supported the graphic design process and the printing of the booklets, as well as academic staff time to support the work. Copies were distributed to memory clinics and post-diagnostic support organizations across Wales from August 2020 onward. Up to October 2023, a total of 8924 copies of the booklet have been requested and disseminated to professionals and people living with dementia across Wales, with some of these being commissioned by local authorities and services (Table 3 provides a breakdown of the distribution in Wales). An electronic copy is also available on the university (dsdc.bangor.ac.uk/products-created) and DEEP (dementiavoices.org.uk) websites, and the booklet was registered with an International Standard Book Number (ISBN 978-1 84220-197-8).

Following the development of the first booklet, the Caban group continued to collaborate and, continuing with the momentum from the development of the first booklet, the group felt they had more personal insights to share. The focus this time was on practical adaptation strategies for day-to-day living with dementia. The group proceeded to collate material for a follow up edition, ‘Knowledge is Power 2: Handy and helpful tips for day-to-day life with dementia’, that includes the following sections: Cooking and eating; Hobbies and activities; Enjoying the outdoors, Household chores; Shopping; Smart devices; and Organization. The bilingual ‘Knowledge is Power 2’ has been available online and as a hard copy since August 2022, with 5466 copies distributed and 4000 of those being commissioned by local authorities and services.

The first booklet was shared with DEEP members in Scotland, who decided to develop an adaptation for Scotland in English and Scottish Gaelic in 2021 [27,28]. Furthermore, an adaptation for England has been developed together with DEEP group members in England and has been published in 2023 [29]. Figure 1 illustrates the ongoing development of the ‘Knowledge is Power’ series.

### 3.5. Feedback

In October 2022, a request was sent out via email and social media requesting feedback on the original booklet using a short survey on Microsoft Forms. It included a basic demographic question (“Are you…” (a person living with dementia/family member/professional/other)) plus three feedback questions: “How useful would you rate the Knowledge is Power booklet?” (rate 1–5 stars); “If we order additional copies is there anything you think we should amend or add to the booklet?” (short sentence responses); and “Any other comments or feedback you would like to share?” (short sentence responses). It yielded 24 responses from professionals working to support people with dementia (*n* = 17), people living with dementia (*n* = 2), family members (*n* = 1) and others (*n* = 4; the chair of a local support group, a volunteer with a local Dementia Friendly Community, a national dementia helpline, and a charity supporting carers). When asked “How useful would you rate the Knowledge is Power booklet”, a mean rating 4.88 out of 5 (SD = 0.34; range: 4–5) was provided.

People who work to support people with dementia highlighted that the booklet is useful, helpful and well-received by those who receive a copy.


*“Booklet has been well received by all our patients and carers”*



*“This is such a great place for useful information with everything together to share and help.”*


It also serves as a valuable starting point for conversations and seeking support:


*“Very simple easy to read book—ideal for giving out to people who want to know where to start and to keep for future reference”*


Respondents were asked whether any amendments should be made to the booklet. Fourteen did not think amendments were needed, with some elaborating “I feel the booklet is just right, if too much information people get bored” and “No I think that the information given is excellent.” Nine respondents made suggestions for minor edits, including adding information about direct payments, how Citizens Advice can help, ensuring the inclusion of telephone numbers alongside web addresses wherever possible, more information for carers, and a section on community transport. The group met in person to discuss the feedback and, subject to agreement and approval from the group, the booklet was revised to incorporate suggestions for minor amendments from the survey and the group members and translated. During the proofreading process, a new title was proposed for the Welsh version: ‘Grym mewn Gwybodaeth’. The new and original titles were put to a poll on social media, with 94% preference for the new Welsh name (*n* = 68 votes). The most recent iteration of the booklet was printed in February 2023, and to make the material more accessible the group began the process of creating an audio version in early 2023.

## 4. Discussion

This paper describes the approach of using human-centered design principles to co-design a resource by people living with dementia to support others living with dementia after diagnosis. By supporting the members of the group to lead the work (through, for example, arranging meetings, collating information for discussion, making revisions, and ensuring appropriate and accessible design), the final product was fit for its intended purpose and tailored according to the most important informational needs of people following a diagnosis in Wales. Furthermore, the success of the booklet is reflected in the positive feedback from those who have used it, its wide demand and distribution across Wales, and the development of further adaptations.

Alzheimer’s Disease International importantly addressed the stigma associated with dementia, highlighting that it “leads to a focus on the ways in which the person is im-paired, rather than on his or her remaining strengths and ability to enjoy many activities and interactions with other people” [30] (p. 10). The co-design work presented in this article counters this harmful stigma about people living with dementia and supports other authors who demonstrate how people living with dementia can not only contribute to society after diagnosis [31] but can spearhead and lead initiatives. One member of the Caban group reflected:


*“So often we are not included. They don’t speak to the people who have dementia to see what works for people with dementia. So, this is showing it really does work.”*
(Caban member)

This work adds to the small number of papers working with people living with dementia to develop informational resources [6,8]. Like Dupuis et al. [6], this work was driven by a desire for accessible and relevant information around the time of diagnosis. However, while Dupuis et al. [6] reported on developing an online self-management tool working as a multidisciplinary team (including people living with dementia) involving a vast and broad amount of useful information that could potentially be used as a ‘one stop shop’ for generic information by anyone in Canada, in this work, a group of people living with dementia directed the volume and content based on personal experience. The group members did not want to include excess information but rather what they felt was the most helpful information required at diagnosis—concise enough to be contained within an A5 booklet and including telephone numbers and uniform resource locators (URLs; for ease of access from the electronic version) where possible. This is perhaps a step beyond the ‘action’ of a leaflet containing generic advice for a local NHS trust produced in Pipon-Young et al. [8], although very little detail is given around the development of the resource in that paper.

### 4.1. Strengths and Limitations

Despite some initial face-to-face meetings with the Caban group and other groups of people living with dementia in developing the content for this work, much of the process of co-design presented in this article took place online during the COVID-19 period of lockdown. We recognize that this could have led to the digital exclusion of some people who may have wished to take part. Nevertheless, every effort was made to ensure that people who wanted to be involved were supported to take part. A strength of this work is that it was funded, which meant that it was possible to purchase and supply tablets to members who did not have access to a computer, as well as the staff time to support the work. Moreover, given that the first COVID-19 lockdown began during this work, meeting on videoconferencing platforms was new to some people and difficulties connecting to Zoom [24] were experienced. However, JR was able to support people to access Zoom [24] over the phone.

We acknowledge some strengths and limitations regarding the input of information towards the development of ‘Knowledge is Power’. Information was predominantly received from members of DEEP groups in North Wales. It is therefore possible that some information may have been missed that could have been contributed by people living in other areas, or by people who are outside of the DEEP network. However, the request for feedback made in late 2022 yielded very few suggestions for revisions, suggesting that the booklet and its contents are appropriate for people Wales-wide. Moreover, ensuring that the booklets are bilingual makes them accessible to people who speak both official languages of Wales, a detail importantly replicated in the Scottish adaptation.

A strength of this work, and an important factor in successful human-centered design, is the relationship between the Caban group members and the academics. The group have built a relationship over time through meeting and working together for several years. This importance of time and developing good relationships is highlighted by Span and colleagues [32], and, as argued by Morbey et al. [33], involvement should be “more than ‘one off’ input”. The friendship, trust and respect between all involved leads to open, honest and meaningful interactions.

### 4.2. Implications for Policy and Practice

The impact of receiving a diagnosis of dementia on the person living with the diagnosis is, understandably, primarily reported as negative [34]. This in turn poses a threat to a person’s resilience [35,36] with negative feelings that the diagnosis is ‘the end’. Alternatively, positive connections with services, support and healthcare professionals which focus on encouraging people to focus on what they can do have the potential to support a person’s resilience. This work has demonstrated the value of adopting human-centered design principles to guides that can support people to live as well as possible following a diagnosis of dementia. The benefit of this work and the resulting resource is that it can be shared at the time of diagnosis. It begins with quotes from people living with dementia and goes on to provide a manageable amount of essential information at a time when people need it.

The development process has been described in such a way that it can, and indeed has been, used by others who wish to develop similar products, e.g., in more local contexts, different countries and languages, or other progressive and terminal conditions. Moreover, feedback on further adaptations through the DEEP network will enable the guides to be accessible in audio formats, broadening accessibility to the information presented in the booklets. The booklets have also prompted discussions on the importance of support provided in the relevant persons’ spoken language rather than assuming that English is the most accessible language for everyone. These considerations are in line with the principles of universal design, which advocate for the “design of products and environments to be usable by all people, to the greatest extent possible, without the need for adaptation or specialized design.” [37].

A formal evaluation of the impact of ‘Knowledge is Power’ would be a logical next step in this work. Ideally, this would include input from all three nations, both from people living with dementia and from those who are distributing the booklets. This should take place once the adaptation for England has been in circulation and had time to reach people living in different regions.

In Wales the importance of co-production is central to policies and strategies relating to dementia, e.g., [4,38]. The Caban group highlighted the issue of an overwhelming amount of written information being provided at diagnosis, something that has been echoed elsewhere [22], and co-designed a solution for the benefit of others in the future. Policy makers in Wales and further afield may wish to incorporate ‘Knowledge is Power’ as a first step in providing information to people newly diagnosed with dementia. This co-design work is consistent with the disability rights movement of “nothing about us without us,” whereby people with lived experience influencing policy and service development is imperative if they are to be designed appropriately. This has already been successfully implemented by two organizations in North Wales who have each commissioned the printing of 2000 copies of ‘Knowledge is Power 1’ and 2000 copies of ‘Knowledge is Power 2’ for their clients.

## 5. Conclusions

In conclusion, the human-centered design approach to resource development presented here provides evidence for the value of this approach in co-designing resources with people living with dementia. Being driven by the target audience ensures that the resource is appropriate and fit for its intended purpose. Furthermore, it demonstrates the importance of including the voices of people with dementia as equal partners to support living as well as possible with dementia.

## Figures and Tables

**Figure 1 ijerph-20-06937-f001:**
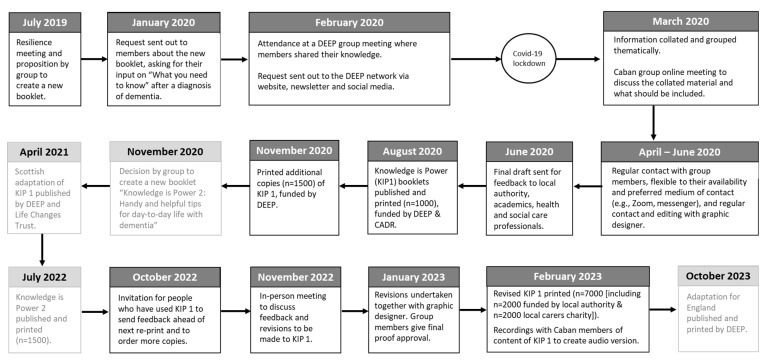
A visual representation of the timeline of development of ‘Knowledge is Power 1’. Note: light-grey boxes illustrate key timepoints in the development of other adaptations.

**Table 1 ijerph-20-06937-t001:** Demographic information about the Caban group.

		People with Dementia	Carers
Sex			
	Male	2	
	Female	3	2
Age			
	Mean	63	39
	Range	51–76	24–54
Marital status			
	Single	1	1
	Married	4	1
Lives with person with dementia/carer			
	Yes	4	2
	No	1	
Person with dementia:			
Dementia diagnosis			
	Alzheimer’s disease	3	
	Mixed Alzheimer’s disease and vascular dementia	2	
Time since diagnosis			
	Mean	5.8 years	
	Range	3–8 years	
Age at diagnosis			
	Mean	57.2	
	Range	48–72	

**Table 2 ijerph-20-06937-t002:** Phases of the ‘Double Diamond’ approach and application by the Caban group.

Phase	Definition	Co-Design by the Caban Group
Discover	This stage focusses on empathizing and understanding the people and the problem to be addressed	A group discussion around what it takes to be resilient while living with dementia and what can affect resilience for people living with dementia
Define	This stage looks at what matters most and how should it be acted upon	Decision by the group to develop a resource (a booklet) to target the needs of people living with dementia at or around the time of diagnosis
Develop	This stage involves developing potential solutions to the problem	Co-designing and creating a booklet utilizing the combined skills and knowledge of the group, facilitated by academics at Bangor University
Deliver	This stage involves delivering the final product to those that it has been created for	Distribution, adaptation for use in other contexts, and the development of a second booklet

**Table 3 ijerph-20-06937-t003:** A breakdown of ‘Knowledge is Power 1’ booklet distribution and requests at each stage.

	First Print 2020 (*n* = 1000)		Reprint 2020 (*n* = 1500)		Reprint 2023 (*n* = 7000)	
	Number of Orders Placed	Total Copies Distributed	Number of Orders Placed	Total Copies Distributed	Number of Orders Placed	Total Copies Distributed
People with dementia	7	44	8	127	8	350
Carer/supporter	4	8	1	7	6	7
Artists and arts organizations	0	0	2	33	4	29
Academics/researchers/students	3	10	4	65	3	251
Health and social care professionals	7	434	14	759	16	975
Care homes and day-care centers	1	20	0	0	1	5
National, devolved and local government	2	119	1	30	4	2165
Organizations representing and supporting PLWD	2	59	2	65	3	140
Third sector (charities)	2	74	3	304	4	2250
Conferences	2	0	1	65	4	252
Unknown	-	232	-	45	-	
		1000		1500		6424

Note: ‘Number of orders placed’ details the number of orders placed by different individuals or organizations; ‘Total copies distributed’ provides the total number of booklets requested and distributed within those orders.

## Data Availability

Data sharing not applicable.

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
