# Peer review of "Knowledge Is Power: Utilizing Human-Centered Design Principles with People Living with Dementia to Co-Design a Resource and Share Knowledge with Peers"

_ijerph, 2023, doi:10.3390/ijerph20206937_

Round 1

Reviewer 1 Report

REVIEWER Comments:

I appreciate the effort the authors have invested in this manuscript, which aims to explore the impact of human-centered design in dementia care. The topic is timely and relevant, and the manuscript provides valuable insights into the subject. Overall, the paper is well-structured and offers a comprehensive overview; however, there are areas that could benefit from further refinement.

Introduction

The introduction is well-written, providing a clear context and rationale for the study. The emphasis on human-centered design and its application in the context of dementia is both relevant and timely. With minor refinements, this introduction sets a strong foundation for the rest of the article.

1.      The introduction provides a clear definition and understanding of human-centered design and action research. However, for readers unfamiliar with these concepts, a more detailed explanation or a brief comparison between the two might be helpful.

2.      The introduction could benefit from a brief statement on the expected outcomes or objectives of the study, giving readers a hint of what to anticipate in the subsequent sections.

Results

1.      The detailed description of the Caban group and its origins provides a solid foundation for understanding the context of the study.

Discussion:

1.      Consider discussing potential limitations of the study, such as potential biases in the co-design process or challenges faced during the development of the resource.

2.      It might be beneficial to provide more details on the potential implications for policy or practice, especially in the context of dementia care.

3.      The discussion could benefit from a brief reflection on future research directions or potential areas of exploration based on the study's findings.Subgroup analysis is essential to differentiate between the impacts on athletes at various levels, and the absence of this facet somewhat limits the depth of the findings. The incorporation of subgroup analysis will cater to a nuanced understanding of the impacts, thus enhancing the study's contribution to the existing knowledge base.

Reviewer 2 Report

Thank you for the opportunity to review this manuscript. Your work is very important and I think it makes a meaningful contribution to the body of literature re co-design and meaningful inclusion of person living with dementia/loved ones in research and health services. The manuscript is very well-written and provides examples of human-centred design principles in practice. The narrative of the processes and flow of actions will be valuable to others interested in pursuing a co-designed project.

I have a few minor suggestions to consider that may strengthen the manuscript for publication.

1.    In your table 1 (demographics), you indicate gender, but sex is a more appropriate label.

2.    On line 124 you note “[see 20]”. It may be helpful to indicate that this is a publication of the findings from the research conducted with the Caban group re resilience and adapting to and managing life with dementia. Similarly, it may be helpful to provide an explanation of what brought the academics/students together with the Caban, initially. This would help to explicate the relationship in place prior to embarking on the codesign of the booklets.

3.    Figure 1 is very helpful in showing the order and flow of events. I think it is helpful in clarifying the two booklets and that work is ongoing. However, within the body of the manuscript (lines 212 – 219) the description of the second booklet development currently reads as somewhat extraneous, or as an add-on. Perhaps explaining that the Caban members and the larger group of academics/students with the Caban group continued to collaborate, that codesigning KP2 continued the group’s actions and momentum would help to frame why the work on KP2 was essential and described within this paper?

4.    Clarifying the group make-up and membership will be helpful; at some places Caban members are referred to, ‘the group’ and ‘the core group’ (lines 250/251). On Lines 95/96, the project is described as led by the Caban group with support from academics. Clarifying the group versus core group versus Caban group may be helpful for readers to understand.

5.    A description of the role that the academics played in the co-design processes is needed. As well, it may be helpful to include brief description of how decisions were made and agreement was achieved/approvals achieved (line 251). These are important processes and details describing the “how” may help others to understand how human-centred design can be used to co-design. Similarly, how did the academics support members of the group’ (line 261).

6.    It seems that this project counters harmful stigma about persons living with dementia – that they are unable to meaningfully engage and contribute to work. This work shows that the opposite is true; persons living with dementia can not only engage, but spearhead and lead initiatives. I think that the discussion could address this important point. How often the group met, and whether the work was funded could add to this. Related to this, as a reader I was curious to know if any challenges were experienced and how the team mitigated these. The conclusion of a paper by Morbey et al. (Trials, 2019) describes that involvement should be more than a ‘one off’ and should instead be meaningful/engaged, then describes how resources impact ability to fully engage.

You provided excellent description of human-centred design and the important of using this in pursuing co-design work. Your succinct description of ethical practices (in absence of formal ethics review/mandated processes) was great.

Overall, a pleasure to read this paper and learn more about how to achieve meaningful co-design with persons living with dementia and their loved ones.

Reviewer 3 Report

The authors present their co-design process to develop a life hack booklet for people living with dementia. The paper is very interesting and the motivation of the presented work is commendable.

- Line 70-77: Authors can also mention the mobile application for early stage dementia:

Chaudhry, B., & Smith, J. (2021). RefineMind: A Mobile App for People with Dementia and Their Caregivers. In The Next Wave of Sociotechnical Design: 16th International Conference on Design Science Research in Information Systems and Technology, DESRIST 2021, Kristiansand, Norway, August 4–6, 2021, Proceedings 16 (pp. 16-21). Springer International Publishing.

- The authors mentioned young onset of dementia, yet the average is 63 years. Why is it important to mention young onset of dementia when the actual sample is quite old and hence study has been done with an older population.

- "Age" is misplaced in Table 1.

- The research ethics part is confusing. Are the authors suggesting that ethical approval for a research can be substituted somehow. If yes, please clarify what authors mean by confirmatory process. On the other hand, without proper ethical approval, data cannot be collected and research cannot be published. If some ethical approval was obtained, the approved protocol must be submitted.

- The authors describe their approach as human-centered co-design approach. My question is can a co-design approach not be human-centered, or is it possible to not use human-centered design principles when using co-design approach. I would recommend just revising the paper so it is more about how co-design approach was implemented using double diamond of design.

- Authors can present some limitations of the work as well as future directions.

Reviewer 4 Report

Thank you for the opportunity to review this important paper. Efforts to integrate patient lived experience into healthcare is so vital and I really applaud and admire the author’s efforts to help those with a diagnosis of dementia living in Wales (and now spreading across the UK). Below are some small suggestions to improve the clarity of the manuscript. They all constitute minor revisions.

1. Line 9: Alzheimer’s Society UK uses “young-onset dementia” with the hyphen between young and onset. (https://www.alzheimers.org.uk/about-dementia/types-dementia/young-onset-dementia)

2. Line 45: “to create and share information with others around the point of a diagnosis of dementia” This could be more clearly stated. E.g., “to create and share information with others on dementia diagnosis and what it entails”

3. Line 66: “names of stages differ” I would add “..names of stages within the design process differ” for clarity. 

4. Line 71: “such as for example” use either “such as” or “for example”, both together are redundant

5. After reading through the introduction, Lines 27-33 seem to be a bit out of place and spoil the flow. I think it would be better to move that after line 77 of the introduction and maybe rework it a little as a research purpose statement. I think this would improve the flow.

6. Line 85: Change “which includes” to “that included”

7. Line 90: I would change the work “mixed” to “a combination of both”

8. Table 2, I think it would be good to have a space (empty line) to separate the four items in the table. 

9. Line 174: “The collated information was grouped thematically and…” Who did the grouping? What themes emerged? I would expand on this a little.

10. Line and 179: “messenger” Change to Messenger

11. Line 182: “The booklet also used the DEEP guidelines for writing dementia friendly information…” The DEEP guidelines were also used throughout the making of the booklet to ensure dementia friendly…”

12. Lines 184-187 “The A5 size was appropriate for accessibility, in that it would be easy to put in a bag and the flip booklet design would enable those receiving the booklet to choose which language they would prefer to access the information.” I would divide this into two sentences/parts. In the first discuss the size, and in the second I think it would be good to give some details about the “flip booklet design” and how it divided the languages in terms of layout. 

13. Lines 197–200: Maybe capitalize the section names.

14. Lines 259-260: “This paper describes the approach of using human-centered design principles to co-design a resource by experts by experience living with dementia to support their peers as 260 they receive a diagnosis.” This sentence is a little bit confusing (underlined part). Please rework it for clarification. 

15. Line 275: I don’t think you can have a semi-colon here after “Canada;” because the sentence clause beginning, “However, while…” isn’t finished.

16. While your paper doesn’t really need a limitations section, I think it would be good, perhaps somewhere in the discussion, if you could discuss any difficulties you encountered in the co-design process and how you dealt with them. I think it would be useful information for people who might try to emulate the fantastic work that you are doing.

17. For clarity, I think it’s good practice to use the oxford comma throughout the manuscript. https://www.oxford-royale.com/articles/oxford-comma/

18. Please add manufacturer's name, city and country for every software (Zoom, Microsoft, Meta, etc.)

Some minor editing would be helpful for clarity, most are indicated in the comments above.

Round 2

Reviewer 3 Report

The authors have revised the contents of the paper based on the reviewers' feedback and now it is much easier to understand. 

please review sentence structures of the newly added sections and make sure they read properly.

Author Response

Please review sentence structures of the newly added sections and make sure they read properly.

Thank you for this suggestion.

We have reviewed the newly added sections and amended sentence structures to make sure they read properly. These can be seen in 'Track Changes' in the revised manuscript.